# Does pre-existing morbidity influences risks and benefits of total hip replacement for osteoarthritis: a prospective study of 6682 patients from linked national datasets in England

Rory Ferguson [1] ,[1] Daniel Prieto-Alhambra [1] ,[2] George Peat [1] ,[3] Antonella Delmestri [1] ,[1] Kelvin P Jordan [1] ,[3] Vicky Y Strauss,[2] Jose Maria Valderas [1] ,[4] Christine Walker,[5] Dahai Yu [1] ,[3] Sion Glyn-Jones,[1] Alan Silman [1] [1]

► http://dx.doi.org/10.1136/bmjopen-2020-046713

For numbered affiliations see end of article.

**Correspondence to**
Professor Alan Silman;
alan.silman@ndorms.ox.ac.uk

## ABSTRACT

Total hip arthroplasty (THA) surgery for elderly people with multimorbidity increases the risk of serious health hazards including mortality. Whether such background morbidity reduces the clinical benefit is less clear.

**Objective** To evaluate how pre-existing health status, using multiple approaches, influences risks of, and quality of life benefits from, THA.

**Setting** Longitudinal record linkage study of a UK sample linking their primary care to their secondary care records.

**Participants** A total of 6682 patients were included, based on the recording of the diagnosis of hip osteoarthritis in a national primary care register and the recording of the receipt of THA in a national secondary care register.

Data were extracted from the primary care register on background health and morbidity status using five different constructs: Charlson Comorbidity Index, Electronic Frailty Index (eFI) and counts of comorbidity disorders (from list of 17), prescribed medications and number of primary care visits prior to recording of THA.

**Outcome measures** (1) Postoperative complications and mortality; (2) postoperative hip pain and function using the Oxford Hip Score (OHS) and health-related quality of life using the EuroQoL (EQ)-5D score.

**Results** Perioperative complication rate was 3.2% and mortality was 0.9%, both increased with worse preoperative health status although this relationship varied depending on the morbidity construct: the eFI showing the strongest relationship but number of visits having no predictive value. By contrast, the benefits were not reduced in those with worse preoperative health, and improvement in both OHS and EQ-5D was observed in all the morbidity categories.

**Conclusions** Independent of preoperative morbidity, THA leads to similar substantial improvements in quality of life. These are offset by an increase in medical complications in some subgroups of patients with high morbidity, depending on the definition used. For most elderly people, their other health disorders should not be a barrier for THA.

## STRENGTHS AND LIMITATIONS OF THIS STUDY

⇒ National sample of older patients with newly reported osteoarthritis of the hip who received a hip replacement.

⇒ Multidimensional approach to assess their concurrent morbidity and health status prior to surgery from the primary care record.

⇒ Linkage to surgical outcomes, including both hazards (mortality and significant postoperative complications) and benefits (postoperative pain, function and quality of life) of THA.

⇒ The challenge of using such routine data sources is to quantify the completeness of recording and accuracy of the data items extracted.

## BACKGROUND

National data on the short-term outcome from total hip arthroplasty (THA) demonstrate a substantial improvement in quality of life.[1] Most recipients are older adults: the median age, for example, in the UK countries reporting to the National Joint Registry is 69 years.[2]

With increasing age, patients are increasingly likely to develop multiple chronic health disorders (multimorbidity) with many patients described as frail.[3–7] Such disorders increase the hazards from surgery[8–10] and may also limit the benefits in quality of life.[11–14] There is in practice an inevitable selection process prior to performing surgery on older adults with multimorbidity, though there is no specific guidance on how these factors should influence surgical decision making, either in UK[15] or USA.[16]

There are challenges in identifying a valid approach to measure the cumulative severity

of all chronic health disorders.[17] Simple counts of the presence of specific chronic diseases, or weighted instruments (eg, Charlson score),[18] typically focus on disorders associated with the hazards especially mortality but exclude those that might impact on quality of life.[19] In a companion paper to this one, we have demonstrated that in current practice in the UK, pre-existing health problems, even when only moderate in effect, do influence the likelihood and timing of THA. The question which then arose is whether this impact on acceptance for surgery appropriately reflects how far these health issues impact on surgical outcome.[20]

Our goal was to identify the impact of multimorbidity and frailty on the risks and benefits of THA. Our specific objectives were to assess, using a number of different approaches to scoring, how multimorbidity influenced the risks and benefits of elective total hip replacement for osteoarthritis. The former outcome was assessed by postoperative complications, length of stay and hospital readmission; and the latter assessed by patient-reported outcome measures (PROMs) covering pain, function and quality of life. To maximise the external validity, the study took advantage of the availability of English national datasets linking primary and secondary care.

## METHODS
### Summary of design
In a longitudinal record linkage study using a national database of primary care records, patients with a newly recorded diagnosis of hip osteoarthritis were identified. Linkage to a national database of secondary care records identified which individuals who had a THA; these individuals constituted the cohort of the study. The primary care record was interrogated to provide information on other health disorders and treatments were used to derive measures of pre-existing multimorbidity and health status at the time of surgery. This secondary care database also provided data on perioperative death and major complications as well as scores of patient outcome. The influence of the scores from these measures on mortality, inpatient complications and readmission within 90 days and postoperative PROMs were then calculated.

### Data sources
We used the UK Clinical Practice Research Datalink (CPRD) which contains the primary care electronic medical records of approximately 4.4 million active patients and representative of the wider UK population.[21] Data are stored with Read codes for diseases that are cross-referenced to the International Classification of Diseases-10.

The Hospital Episode Statistics (HES) dataset contains all patient attendances at National Health Service (NHS) hospitals in England, and covers episodes of care. HES records patient demographic data, diagnoses and procedures and is linked to national death certificate data and these data were provided anonymised. HES has also

been linked since 2009 to the PROMs database gathering outcome data of NHS-funded THA, both preoperatively and at 6 months postoperatively.[22]

### Study population
All patients in the CPRD with a Read code for hip osteoarthritis from 1 January 1995, until 31 March 2014, and aged ≥65 years at the time of diagnosis, were identified (ISAC protocol number 17_024R). The validation of those Read codes for hip osteoarthritis are described elsewhere.[23] Those patients who underwent elective primary THA, identified from HES, were included in the analysis. Operations were identified from the Classification of Surgical Operations and Procedures-4 codes.

The following measures of multimorbidity prior to the date of surgery were extracted from the accumulated primary care record. These were (1) Charlson Comorbidity Index (CCI)—developed to predict 1 year mortality based on the presence of specified chronic diseases,[18] (2) count of the 17 chronic diseases listed in the NHS Quality Outcomes Framework[24], (3) electronic Frailty Index (eFI), a score based on the cumulative deficit model of frailty and validated against mortality, hospitalisation and nursing home admission.[25] We also used two measures of the burden of care associated with the presence of chronic conditions. Thus, within the 12 months prior to surgery, we extracted the number of (1) different prescribed medications and (2) primary care contacts for any reason.

### Outcomes
All subjects had the following outcomes, in addition to death, extracted from the linked HES data within 90 days of THA: venous thromboembolism, myocardial infarction, stroke, anaemia, lower respiratory tract infection, urinary tract infection and wound infection. Also extracted were length of stay following surgery and readmission after discharge within 90 days. The national PROMs database provided data since 2009 on (1) Oxford Hip Score (OHS) as a measure of hip function and (2) EuroQoL (EQ)-5D Index as a measure of quality of life at 6 months postoperatively on all patients receiving a THA—information on the appropriate interpretation of these scores is provided in the online supplemental appendix. Although all patients are invited to complete the PROMs questionnaires preoperatively and at 6 months postoperatively, completion rates vary. Further some respondents delay completing their postoperative scores beyond the 6 months.[22]

### Statistical analysis
The scores for each of the five measures of multimorbidity were split into four categories. The eFI has four categories defined by the developers: fit (0–4), mild frailty (5–8), moderate frailty (9–12) and severe frailty (>13). The actual CCI scores were used to split into categories of score of 0, 1, 2 and >3. Based on the distribution of the data, the counts of number of chronic diseases were also

split in categories of score of 0, 1, 2 and >3. The categories for the number of medications prescribed and of primary care visits were derived from the observed distributions of the actual data, aiming for equal-sized quarters: with medication counts divided into scores of 0–4, 5–7, 8–12 and >13 and number of primary care visits into 0–7, 8–11, 12–17 and >18. The lowest category for each score was used as the referent category in the analysis.

Logistic regression methods were used to calculate the unadjusted odds of a complication or readmission to hospital by multimorbidity category. Poisson regression methods were used to calculate the unadjusted difference in length of stay by multimorbidity category. Linear regression methods were used to calculate the difference in postoperative OHS and EQ-5D score, adjusted for preoperative score, by multimorbidity category. All models were then adjusted for age at surgery, sex, region of the UK and calendar year of total hip replacement.

### Patient and public Involvement

This research was funded by the National Institute for Health Research Research for Patient Benefit scheme which has an absolute requirement that there is patient and public involvement in all relevant stages of the research. The research question had been originally raised in a 'Priority Setting Partnership for Priorities for Research in Hip and Knee Arthroplasty' and this was followed by a survey of members of the Keele Patient and Public Involvement panel: Research Users Group (RUG). We then tested the suggested questions with the group and received very positive feedback. One of the RUG members (CW), herself a patient with direct experience of the target of the research, then became an active member of the research team, participated in all the meetings and advised on the design including the questions that should be asked during the analysis stage. As part of the dissemination phase, Keele Patient and Public Involvement group organised a round-table event attended by the lead authors where the results were discussed and guidance given on how those members of the public who were present wished to see disseminated.

### RESULTS
### Baseline data

The demographic characteristics of the study participants are shown in table 1. There were 6682 patients with a code for hip osteoarthritis on the primary care record who after linkage to the HES data, had a THA within the follow-up period from April 1997 to March 2014. The mean age at surgery was 76 years (SD 6).

Preoperative scores for the four multimorbidity measures derived are shown in table 2. Only 20% had a CCI >1 perhaps reflecting the selection of patients for surgery. Based on the eFI classification, 34% were classified as 'mildly frail' and 6.5% 'moderately or severely frail'. Around 1 in 6 had three or more chronic disorders recorded. Over 40% had been prescribed at least eight

| Table 1 | Demographic characteristics of cohort |
|---|---|
| **Gender** | **N (%)** |
| Female | 4090 (61.2) |
| **Age (years)** | |
| 65–69 | 1294 (19.4) |
| 70–74 | 1976 (29.6) |
| 75–79 | 1736 (26.0) |
| 80–84 | 1095 (16.4) |
| 85–89 | 465 (7.0) |
| >90 | 116 (1.7) |
| **Body mass index** | |
| Underweight (<18.5) | 54 (1.0) |
| Normal (18.5–24.9) | 1615 (28.8) |
| Overweight (25–29.9) | 2444 (43.6) |
| Obese (>30) | 1497 (26.7) |
| Missing | 1072 |
| **Index of multiple deprivation** | |
| 1 (Affluent) | 1076 (25.5) |
| 2 | 1718 (25.7) |
| 3 | 1539 (23.0) |
| 4 | 1144 (17.1) |
| 5 (Deprived) | 573 (8.6) |
| Missing | 2 |

medications, and a similar proportion had >12 primary care visits, in the previous 12 months.

### Outcomes

Data on complications, length of stay and readmission to hospital were available on all patients (as linkage to HES was a criterion for study entry). As mentioned above, data on PROMs were only available on a subset of patients who completed the questionnaires (figure 1). In total, preoperative and postoperative OHS were available on 1402 patients, and preoperative and postoperative EQ-5D Index were available on 1285 patients. As a test of bias, we compared the multimorbidity measures between those with and without PROMs (online supplemental table 1) which, while showing no substantive shifts across all the measures, found those who did not provide PROMs were more likely to have worse multimorbidity and also reside in more socioeconomically deprived areas.

### Association between multimorbidity measures and adverse outcomes

In all, 216 (3.2%) patients suffered a postoperative complication including thromboembolism, myocardial infarction, urine and respiratory infections and wound infection, and 57 (0.9%) patients died, within 90 days of surgery (table 3). The rate of all-cause readmission to hospital within 90 days of surgery was 11%.

**Table 2** Baseline data on measures of preoperative health status (multimorbidity, frailty and morbidity burden)

| Charlson Comorbidity Index | |
|---|---|
| 0 | 4490 (67.2) |
| 1 | 682 (10.2) |
| 2 | 880 (13.2) |
| >3 | 630 (9.4) |
| **Electronic Frailty Index** | |
| 0–4 | 4184 (62.6) |
| 5–8 | 2064 (30.9) |
| 9–12 | 393 (5.9) |
| >13 | 41 (0.6) |
| **No. of comorbid diseases** | |
| 0 | 1901 (28.5) |
| 1 | 2286 (34.2) |
| 2 | 1430 (21.4) |
| >3 | 1065 (15.9) |
| **No. of medications prescribed** | |
| 0–4 | 1963 (29.4) |
| 5–7 | 1832 (27.4) |
| 8–12 | 1992 (29.8) |
| >13 | 895 (13.4) |
| **No. of contacts with primary care** | |
| 0–7 | 2167 (32.4) |
| 8–11 | 1598 (23.9) |
| 12–17 | 1528 (22.9) |
| >18 | 1389 (20.8) |

The impact of multimorbidity scores across all these medical complications are shown in table 4. The overall adverse event rate was only modestly increased with

**Table 3** Complications, length of stay and readmission to hospital within 90 days

| | N (%) |
|---|---|
| Any medical complication* | 216 (3.2) |
| Death | 57 (0.9) |
| Myocardial infarction | 20 (0.3) |
| Venous thromboembolism | 54 (0.8) |
| Stroke | 8 (0.1) |
| Anaemia | 12 (0.2) |
| Respiratory tract infection | 28 (0.4) |
| Urinary tract infection | 12 (0.2) |
| Wound infection | 37 (0.6) |
| Length of stay (median, IQR) | 6 (4–9) |
| Readmission to hospital | 727 (10.9) |

*Any medical complication includes death, myocardial infarction, venous thromboembolism, stroke, anaemia, respiratory tract infection, urinary tract infection and wound infection.

increasing levels of multimorbidity relative to the lowest scoring group. These increases in risks were attenuated after adjustment for age and the other possible confounders. Those in highest frailty group were the most predictive of a medical complication. By contrast, the other measures of multimorbidity were only weakly predictive; with some measures such as the number of primary care visits showing no trend. We analysed the individual complications separately (data not shown). The numbers with any of the individual complications were small but there were no substantial difference between the risks of individual complications in their relationship with the multimorbidity scores.

The data on mortality are shown in table 5. The mortality risk in the least healthy stratum ranged from 0.7% for the group with the highest number of primary

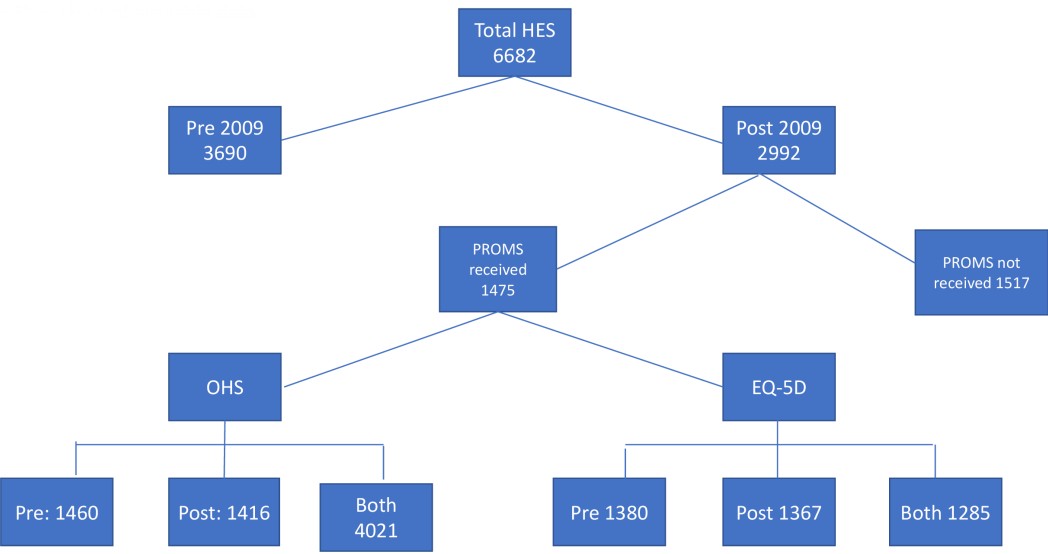

**Figure 1** Flow chart of data available for analysis. HES, Hospital Episode Statistics; PROMs, patient-reported outcome measures.

**Table 4** Association between preoperative morbidity status and 90-day medical complications

| | No. at risk | Events | % Events | Unadjusted OR | Adjusted OR* |
|---|---|---|---|---|---|
| Charlson Comorbidity Index | | | | | |
| 0 | 4490 | 120 | 2.67 | Ref | Ref |
| 1 | 682 | 27 | 3.96 | 1.50 (0.98 to 2.30) | 1.43 (0.93 to 2.20) |
| 2 | 880 | 43 | 4.89 | 1.87 (1.31 to 2.67) | 1.78 (1.24 to 2.56) |
| >3 | 630 | 26 | 4.13 | 1.57 (1.02 to 2.42) | 1.46 (0.94 to 2.26) |
| Electronic Frailty Index | | | | | |
| 0–4 | 4184 | 112 | 2.68 | Ref | Ref |
| 5–8 | 2064 | 75 | 3.63 | 1.37 (1.02 to 1.85) | 1.31 (0.96 to 1.78) |
| 9–12 | 393 | 26 | 6.62 | 2.58 (1.66 to 4.00) | 2.26 (1.42 to 3.62) |
| >13 | 41 | <5† | | | |
| Comorbid diseases | | | | | |
| 0 | 1901 | 53 | 2.79 | Ref | Ref |
| 1 | 2286 | 53 | 2.32 | 0.83 (0.56 to 1.22) | 0.80 (0.54 to 1.18) |
| 2 | 1430 | 57 | 3.99 | 1.45 (0.99 to 2.12) | 1.41 (0.96 to 2.08) |
| >3 | 1065 | 53 | 4.98 | 1.83 (1.24 to 2.69) | 1.74 (1.16 to 2.61) |
| Prescriptions | | | | | |
| 0–4 | 1963 | 49 | 3.09 | Ref | Ref |
| 5–7 | 1832 | 55 | 2.75 | 1.21 (0.82 to 1.79) | 1.17 (0.79 to 1.73) |
| 8–12 | 1992 | 68 | 3.53 | 1.38 (0.95 to 2.00) | 1.28 (0.88 to 1.87) |
| >13 | 895 | 33 | 3.67 | 2.02 (1.33 to 3.06) | 1.85 (1.21 to 2.83) |
| Primary care contacts | | | | | |
| 0–7 | 2167 | 67 | 3.09 | Ref | Ref |
| 8–11 | 1598 | 44 | 2.75 | 0.89 (0.60 to 1.31) | 0.85 (0.58 to 1.25) |
| 12–17 | 1528 | 54 | 3.53 | 1.15 (0.80 to 1.65) | 1.09 (0.76 to 1.58) |
| >18 | 1389 | 51 | 3.67 | 1.19 (0.82 to 1.73) | 1.07 (0.74 to 1.57) |
| Total | 6682 | 216 | | | |

*Adjusted for age, gender, region and calendar year of surgery.
†The rules set for approval of use of CPRD data require that authors are not allowed to indicate the number of events in any category if they are <5.
CPRD, Clinical Practice Research Datalink.

care contacts to 2.3% for those with an eFI of ≥9. There was an inconsistent association between death and Charlson score but the number of events in those with a Charlson score >2 were too small for useful analysis. Neither the groups with the number of prescribed medications nor of primary care contacts predicted mortality. The measure with the greatest predictive power of death was the eFI. After adjustment for age and gender, those in the two highest frailty groups had a 2.5-fold increased mortality risk. The overall number of deaths though was modest and the resulting small numbers when stratified by preoperative morbidity category, limits the precision of these estimates.

The impact of multimorbidity scores on length of stay were also analysed. Detailed data are not shown but in brief for all the multimorbidity measures, apart from the eFI, there was no impact on length of stay. The median length of stays in the lowest and highest categories of multimorbidity were: 6 and 7 days for Charlson score,

6 and 8 days for eFI score, 6 and 7 days for the count of chronic diseases, and 6 and 7 days for the count of medications prescribed.

We have also examined the impact of these preoperative health measures on two indirect measures of complications: readmission rates and length of stay (LOS). Overall, there were 786 patients readmitted for any reason over the next 90 days. Risks were related to level of preoperative health with those in the worst health categories having a twofold increased risk compared with those in in the healthiest group. The results were broadly similar independent of the scoring approach (online supplemental table 2). As regards LOS, the differences between grades of prior ill health were modest, with the median and interquartile ranges almost identical. The eFI was a slight outlier where those in the highest two categories had a median LOS 2 days higher than those with lesser frailty (online supplemental table 3).

**Table 5** Association between preoperative multimorbidity and 90-day mortality

| | No. at risk | Events | % Events | Unadjusted OR | Adjusted OR* |
|---|---|---|---|---|---|
| Charlson Comorbidity Index | | | | | |
| 0 | 4490 | 32 | 0.71 | Ref | Ref |
| 1 | 682 | 6 | 0.88 | 1.24 (0.52 to 2.97) | 1.09 (0.45 to 2.65) |
| 2 | 880 | 13 | 1.48 | 2.09 (1.09 to 4.00) | 1.85 (0.95 to 3.60) |
| >3 | 630 | 6 | 0.95 | 1.34 (0.56 to 3.22) | 1.12 (0.46 to 2.73) |
| Electronic Frailty Index | | | | | |
| 0–4 | 4184 | 27 | 0.65 | Ref | Ref |
| 5–8 | 2064 | 20 | 0.97 | 1.51 (0.84 to 2.69) | 1.33 (0.72 to 2.44) |
| 9–12 | 393 | 10 | 2.54 | 4.02 (1.93 to 8.37) | 2.78 (1.24 to 6.23) |
| >13 | 41 | 0 | 0 | – | – |
| Comorbid diseases | | | | | |
| 0 | 1901 | 14 | 0.74 | Ref | Ref |
| 1 | 2286 | 10 | 0.44 | 0.59 (0.26 to 1.34) | 0.58 (0.26 to 1.33) |
| 2 | 1430 | 16 | 1.12 | 1.53 (0.74 to 3.14) | 1.50 (0.71 to 3.16) |
| >3 | 1065 | 17 | 1.60 | 2.19 (1.07 to 4.45) | 2.12 (0.99 to 4.51) |
| Prescriptions | | | | | |
| 0–4 | 1963 | 11 | 0.56 | Ref | Ref |
| 5–7 | 1832 | 19 | 1.04 | 1.86 (0.88 to 3.92) | 1.80 (0.85 to 3.82) |
| 8–12 | 1992 | 18 | 0.90 | 1.62 (0.76 to 3.43) | 1.39 (0.64 to 2.98) |
| >13 | 895 | 9 | 1.01 | 1.80 (0.74 to 4.37) | 1.51 (0.61 to 3.75) |
| Primary care contacts | | | | | |
| 0–7 | 2167 | 20 | 0.92 | Ref | Ref |
| 8–11 | 1598 | 13 | 0.81 | 0.88 (0.44 to 1.78) | 0.81 (0.40 to 1.65) |
| 12–17 | 1528 | 15 | 0.98 | 1.06 (0.54 to 2.09) | 0.90 (0.45 to 1.78) |
| >18 | 1389 | 9 | 0.65 | 0.70 (0.32 to 1.54) | 0.53 (0.24 to 1.19) |
| Total | 6682 | 57 | | | |

*Adjusted for age, gender, region and calendar year of surgery.

## Association between multimorbidity measures and quality of life

Across all patients, there was an overall marked improvement in both the specific (OHS) and the generic (EQ-5D) patient outcomes. The summary outcome data are shown in table 6, confirming substantial improvement at 6 months in both the OHS and EQ-5D across the population studied. The mean improvement in OHS of around 20 did not vary importantly between the different levels of any of the approaches to assessing the preoperative morbidity (table 7). For each of our measures of multimorbidity, the preoperative OHS was lower with increasing score. What was of note was the similarity in the relative improvement in the postoperative OHS seen across the board for all measures of multimorbidity. Typically mean preoperative scores in the mid-to-late teens reached scores in the high 30s and 40s; the highest score achievable is 48. Thus, surgery led to a marked improvement, not attenuated by the severity of the preoperative general health.

A similar analysis was undertaken on EQ-5D (table 8). As expected, those with a higher preoperative morbidity had a lower preoperative EQ-5D, reflecting the impact of the patients' other comorbid disorders on their overall quality of life. However, the 6-month incremental improvement in EQ-5D was almost identical across all categories of multimorbidity, independent of the method of multimorbidity scoring. The actual EQ-5D scores were

**Table 6** Patient-reported outcome measures

| | N | Mean (SD) |
|---|---|---|
| Oxford Hip Score | | |
| Preoperative | 1460 | 17.9 (7.9) |
| Postoperative | 1416 | 39.3 (8.2) |
| Change | 1402 | 21.3 (9.7) |
| EuroQoL-5D Index | | |
| Preoperative | 1380 | 0.35 (0.31) |
| Postoperative | 1367 | 0.80 (0.23) |
| Change | 1285 | 0.45 (34) |

**Table 7** Postoperative Oxford Hip Score by multimorbidity

| | No. | Preoperative | Postoperative | Change | Unadjusted difference | Adjusted* |
|---|---|---|---|---|---|---|
| **Charlson Comorbidity Index** | | | | | | |
| 0 | 960 | 18.5 (8.0) | 39.9 (8.0) | 21.4 (9.6) | Ref | Ref |
| 1 | 113 | 17.6 (8.4) | 37.5 (9.7) | 20.1 (10.5) | −2.0 (−3.6 to −0.5) | −2.0 (−3.6 to −0.5) |
| 2 | 199 | 16.0 (7.5) | 37.8 (8.3) | 21.9 (9.4) | −1.4 (−2.6 to −0.2) | −1.0 (−2.2 to 0.2) |
| >3 | 130 | 16.8 (7.1) | 38.5 (8.1) | 21.5 (10.0) | −1.15 (−2.5 to 0.4) | −0.60 (−2.0 to 0.8) |
| **EFI** | | | | | | |
| 0–4 | 798 | 18.8 (7.9) | 40.7 (7.5) | 21.8 (9.4) | Ref | Ref |
| 5–8 | 511 | 17.3 (7.9) | 37.7 (8.8) | 20.5 (10.2) | −2.5 (−3.4 to −1.7) | −2.0 (−2.9 to −1.2) |
| 9–12 | 86 | 14.6 (7.0) | 35.8 (8.1) | 21.4 (8.8) | −3.6 (−5.4 to −1.9) | −2.9 (−4.7 to −1.1) |
| >13 | 7 | 14.7 (8.2) | 39.1 (10.4) | 24.4 (8.7) | −0.45 (−6.2 to 5.3) | 1.0 (−4.8 to 6.7) |
| **Comorbid diseases** | | | | | | |
| 0 | 358 | 18.9 (7.8) | 40.7 (7.9) | 21.7 (9.9) | Ref | Ref |
| 1 | 481 | 18.6 (8.1) | 39.9 (7.9) | 21.3 (9.6) | −0.7 (−1.8 to 0.4) | −0.4 (−1.5 to 0.6) |
| 2 | 324 | 17.6 (7.9) | 38.7 (8.2) | 21.0 (9.6) | −1.6 (−2.8 to −0.4) | −1.2 (−2.3 to 0.1) |
| >3 | 239 | 15.7 (7.5) | 36.9 (8.9) | 21.2 (9.7) | −3.0 (−4.3 to −1.7) | −2.3 (−3.6 to −1.0) |
| **Prescriptions** | | | | | | |
| 0–4 | 407 | 19.9 (7.4) | 41.4 (7.1) | 21.4 (9.3) | Ref | Ref |
| 5–7 | 383 | 19.4 (8.1) | 40.4 (7.4) | 20.9 (9.5) | −0.8 (−1.9 to 0.3) | −0.5 (−1.6 to 0.6) |
| 8–12 | 406 | 16.6 (7.7) | 38.2 (8.5) | 21.7 (9.9) | −2.3 (−3.4 to −1.2) | −2.0 (−3.0 to −0.9) |
| >13 | 206 | 14.1 (7.3) | 35.4 (9.2) | 21.3 (10.3) | −4.6 (−5.9 to −3.2) | −4.0 (−5.3 to −2.4) |
| **Primary care contacts** | | | | | | |
| 0–7 | 2167 | 18.9 (7.8) | 39.9 (7.8) | 21.0 (8.9) | Ref | Ref |
| 8–11 | 1598 | 18.3 (8.1) | 40.1 (7.8) | 21.7 (10.0) | 0.3 (−0.8 to 1.5) | 0.6 (−0.6 to 1.7) |
| 12–17 | 1528 | 17.8 (7.8) | 39.4 (8.1) | 21.7 (9.9) | −0.1 (−1.2 to 1.0) | 0.4 (−0.7 to 1.5) |
| >18 | 1389 | 16.1 (9.1) | 37.2 (9.2) | 21.1 (10.3) | −2.0 (−3.2 to −0.8) | −1.5 (−2.7 to −0.3) |
| Total | 1402 | 1460 | 1416 | 1402 | | |

*Adjusted for age, gender, region and calendar year of surgery.

moderately lower in the highest categories of multimorbidity but the relative improvement was the same. Regression analyses, both unadjusted and adjusted for age and gender showed that the postoperative scores, adjusted for the preoperative scores, were not importantly different (see online supplemental appendix 1) for any of the multimorbidity scores compared with the lowest score in each grading system.

## DISCUSSION
Although previous studies have demonstrated a link between same indices of serious morbidity, such as the Charlson score and subsequent long-term mortality[19], this is the first study to comprehensively assess the impact of multiple approaches to preoperative health measures, on the broad range of risks and benefits of THA. We have demonstrated that there is a link between worse preoperative health status with risk of death and of clinically significant medical event rates, although the size and the steepness of any trend varied between the different approaches to assessing health. There is no single obvious

measure which captures all the relevant dimensions. Of interest was that these approaches in general did not yield consistent differences in these results, although the number of primary care contacts was 'an outlier' in this respect and would not appear to be useful as a measure of health status.

The other important result was that THA is associated with a substantial improvement in quality of life, independent of level of pre-existing multimorbidity. Indeed, this was true for all the approaches used to assess health. This consistency in the level of benefit is thus in contrast to the variable impact between these approaches on complications of surgery.

There are some limitations of these data. First, we have not considered the impact of different types of surgery and anaesthesia together with other prophylactic interventions to reduce the hazards from surgery in this group of patients. For example, the routine use of the use of thromboprophylaxis should reduce the risk of thromboembolism even in those with the highest multimorbidity. Such interventions would not, though, explain the

**Table 8** Postoperative EuroQoL-5D by multimorbidity

| | No. | Preoperative | Postoperative | Change | Unadjusted difference | Adjusted difference* |
|---|---|---|---|---|---|---|
| **Charlson Comorbidity Index** | | | | | | |
| 0 | 880 | 0.37 (0.31) | 0.82 (0.22) | 0.45 (0.34) | Ref | Ref |
| 1 | 108 | 0.33 (0.32) | 0.77 (0.25) | 0.44 (0.33) | −0.036 (−0.080 to 0.007) | −0.037 (−0.080 to 0.007) |
| 2 | 178 | 0.29 (0.31) | 0.76 (0.22) | 0.48 (0.34) | −0.038 (−0.073 to −0.002) | −0.028 (−0.064 to 0.007) |
| >3 | 119 | 0.33 (0.29) | 0.74 (0.24) | 0.40 (0.33) | −0.072 (−0.114 to −0.030) | −0.060 (−0.102 to −0.018) |
| **Electronic Frailty Index** | | | | | | |
| 0–4 | 735 | 0.39 (0.31) | 0.83 (0.20) | 0.45 (0.33) | Ref | Ref |
| 5–8 | 464 | 0.32 (0.31) | 0.76 (0.23) | 0.45 (0.36) | −0.059 (−0.084 to −0.033) | −0.048 (−0.073 to −0.022) |
| 9–12 | 79 | 0.20 (0.29) | 0.69 (0.28) | 0.47 (0.34) | −0.141 (−0.192 to −0.090) | −0.126 (−0.178 to −0.074) |
| >13 | 7 | 0.20 (0.35) | 0.78 (0.35) | 0.59 (0.32) | −0.021 (−0.184 to 0.141) | .004 (−0.157 to 0.166) |
| **Comorbid dis** | | | | | | |
| 0 | 330 | 0.39 (0.31) | 0.85 (0.19) | 0.46 (0.34) | Ref | Ref |
| 1 | 429 | 0.38 (0.31) | 0.81 (0.20) | 0.43 (0.32) | −0.037 (−0.068 to −0.005) | −0.030 (−0.062 to 0.001) |
| 2 | 306 | 0.33 (0.31) | 0.78 (0.23) | 0.45 (0.35) | −0.060 (−0.094 to −0.026) | −0.049 (−0.084 to −0.015) |
| >3 | 220 | 0.26 (0.32) | 0.71 (0.28) | 0.45 (0.36) | −0.122 (−0.159 to −0.084) | −0.105 (−0.143 to −0.067) |
| **Prescriptions** | | | | | | |
| 0–4 | 378 | 0.42 (0.30) | 0.85 (0.20) | 0.44 (0.34) | Ref | Ref |
| 5–7 | 351 | 0.41 (0.30) | 0.83 (0.20) | 0.42 (0.32) | −0.023 (−0.055 to 0.008) | −0.062 (−0.062 to 0.001) |
| 8–12 | 372 | 0.30 (0.30) | 0.77 (0.22) | 0.47 (0.34) | −0.071 (−0.103 to −0.040) | −0.084 (−0.084 to −0.015) |
| >13 | 184 | 0.19 (0.31) | 0.69 (0.28) | 0.50 (0.37) | −0.139 (−0.178 to −0.099) | −0.143 (−0.143 to −0.067) |
| **Primary care contacts** | | | | | | |
| 0–7 | 2167 | 0.39 (0.31) | 0.82 (0.22) | 0.44 (0.33) | Ref | Ref |
| 8–11 | 1598 | 0.36 (0.30) | 0.83 (0.21) | 0.46 (0.34) | 0.003 (−0.029 to 0.036) | .007 (−0.025 to 0.039) |
| 12–17 | 1528 | 0.36 (0.31) | 0.80 (0.21) | 0.44 (0.35) | −0.018 (−0.049 to 0.013) | −0.010 (−0.042 to 0.021) |
| >18 | 1389 | 0.26 (0.32) | 0.72 (0.26) | 0.47 (0.35) | −0.079 (−0.114 to −0.044) | −0.070 (−0.106 to −0.034) |
| Total | 1285 | 1380 | 1367 | 1285 | | |

*Adjusted for age, gender, region and calendar year of surgery.

observation that improvement in patient outcome is not materially affected by pre-existing morbidity.

Second, inevitably in studies of this type, there is a reliance on the completeness, and quality diagnostic accuracy in the data sets available. We relied on the quality of the CPRD to provide information on the diagnosis of osteoarthritis of the hip and of the multimorbidity scores. As regards the former, a validation study using radiological and clinical information from a subsample of this cohort, described elsewhere, confirmed the diagnosis of hip osteoarthritis in 80%.[20] As regards the latter there are many reports on the validity of the national datasets used for deriving multimorbidity information.[21 26]

There is also a similar reliance on the quality of the outcome data from HES. Although data from HES are widely accepted as being sufficiently valid[27] we were concerned that this data source captured a high proportion of these relevant outcome events with minimal misclassification. As a test of the quality of the outcomes data, we therefore compared our results on postoperative hazard rates with other published data to determine whether such rates were broadly similar. There are limitations in such comparisons, but this analysis suggested that the observed rates of clinically significant events in the population studied did accord with those published from other countries. For example, our VTE incidence was 2.1%, published population data suggest a range of 0.7%–3.9%.[28–30] Similar comparisons for MI were as follows: this study 0.4%, literature 0.4%–0.6%[31–33]; wound infection present study 1.7%, literature 2.2%–2.9%.[34]

Readmission rate is also useful as an indirect measure of complications. Indeed, the impact of both being in the most frail category of eFI and number of prescriptions was similar on the rates of both readmissions (online supplemental table 1) to that seen on complications (table 4). The agreement was less observed with primary care contacts which reflects health seeking behaviours as well as morbidity.

Preoperative health status had little effect on LOS, perhaps surprisingly but this may be complex to unpack given differences in anaesthetic procedure. We specifically studied multiple ways of measuring pre-existing

morbidity and its impact on health, given there is no single 'best way of measuring morbidity, and all such scores are measures of an underlying, but hard-to-measure, construct. This resulted in multiple estimates of effect. We were not attempting to prove statistical significance by finding one or more measure that was useful. More, by using multiple options, we covered as far as possible the different suggested approaches to ascertaining multimorbidity. A detailed statistical analysis identifying the concordance between these measures was outside the scope of this study, we were not attempting to pool the data to provide a single measure of health. Obviously, these different measures will overlap in the patients classified. One of the strengths of this study was that the measures did not yield always similar relationships, especially with the risk studied. This is not, however, surprising as they were measuring different constructs. Despite such differences, none of the scoring systems had substantial effects on outcome. One construct of patient health that was not collected from these routine data is the patient's perception of their own status. Others have shown for example that EQ-5D, measured prior to surgery, is associated with longer term mortality.[35] One problem with using such measures measured at a single point of time is that they fluctuate more than the longer interval used in this study to capture these primary-care recorded events.

There are other variables that might be confounders which have not been addressed in the analyses presented which include socioeconomic status, body mass index and cigarette smoking. We specifically did not adjust for the possible confounding effects of these variables in part because of the quality and completeness of the data. More importantly, we were concerned that the effect of these variables would most likely be mediated by their association with other comorbid disorders and interpretation of such adjusted results would be difficult.

The data on benefits were only collected at 6 months postoperatively. We cannot comment on the long-term impact of these preoperative measures on quality of life. This is not without challenges given the increasing likelihood over time that there may be other changes in health that will impact on overall benefit.

Finally, the patients studied in this cohort were inevitably those who underwent surgery. Our comparisons therefore were between those in the different morbidity categories for whom a decision had been made that surgery was appropriate. We could not address whether, *within each such category*, there were selective differences in surgical decision making that might be associated with the outcomes measured. One reasonable unmeasured confounder relates to prior perception of benefit. Thus, the individuals with major health issues who agree to surgery might have a higher expectation of outcome. Such individuals might then score higher on their self-reported outcomes, than otherwise similar patients who did not have surgery.

This study aimed to investigate the influence of common long-term disorders in the population and their influence on the risks and benefits from hip surgery. It is accepted that those at greatest risk from anaesthesia and major surgery should be identified, especially for an elective procedure such as THA. Preoperative morbidity does increase the mortality risk.[8 10 36] By contrast, there are very little data on how the co-existing health problems affect the clinical benefits. Indeed, we have shown in a recent analysis that those patients who have undergone THA in the highest American Society of Anesthesiologists (ASA)[37] grade (≥III) are at increased risk of revision surgery,[38] a result that is not unexpected. ASA grades while routinely collected for example in national joint registers, are not extractable from primary care data systems. The underlying concern behind undertaking the current study was that, especially in older individuals, the lack of an evidence base weighing up the risks and benefits from surgery, leads to patients, their families and healthcare professionals, accepting that surgery should be avoided. The current study suggests that the increased risks are modest but the benefits remain substantial.

As part of the study design, we have been able to examine a number of different approaches to assessing multimorbidity in primary care patients. Indeed, that setting with the complete medical record provides the only practical one for scoring multimorbidity of patients in clinical practice. The eFI, a recently developed tool for assessing frailty from the primary care record,[25] is achieving widespread usage in the UK and elsewhere for classifying frailty in elderly patients. High eFI scores are associated with substantial increased risks of mortality, unplanned hospital admission and the need for long term care.[25] It is thus interesting that within the current dataset the differences in eFI class were, if anything more predictive of outcomes than the other approaches. It should be noted though that the proportion of the population selected for surgery in the highest eFI category was small. Furthermore, even a high eFI class on our data, should not, of itself, be a barrier to surgery.

We conclude that in older patients requiring a hip replacement, unless they are in the small group whose anaesthetic risk precludes safe surgery, there is no evidence that their other long-term disorders should be a barrier to this life-changing elective procedure.

**Author affiliations**

[1]Nuffield Department of Orthopaedics, Rheumatology, and Musculoskeletal Sciences, University of Oxford, Oxford, UK
[2]Centre for Statistics in Medicine, Nuffield Department of Orthopaedics, Rheumatology, and Musculoskeletal Sciences, University of Oxford, Oxford, UK
[3]Primary Care Centre Versus Arthritis, School of Medicine, Keele University, Keele, UK
[4]Health Services and Policy Research Group, Medical School, University of Exeter, Exeter, UK
[5]Research User Group, Primary Care Centre Versus Arthritis, School of Medicine, Keele University, Keele, UK

**Acknowledgements** We acknowledge Professor Andy Judge for advice on statistical analysis at the earliest stages of this project . We are also grateful to Professor Andrew Clegg and Stephen Pye for guidance in the use of the electronic frailty index. Dr John Griffiths and Professor Nigel Arden, provided very helpful

advice at the design stage on the challenges of anaesthetic assessment and on studying osteoarthritis from an epidemiological perspective. We are grateful to public members of the Research Users Group at Keele University for their helpful input especially in the interpretation of the results.

**Contributors** The study was conceived by AS with SG-J and DP-A and detailed protocol produced by these authors with input from GP, KPJ, JMV and DY. AD was responsible for extracting, preparing and ensuring the quality of the datasets for analysis; analysis was undertaken by RF supported by VS, DP-A and AS, CW provided patient input into the design, approach to analysis and interpretation of the data. All authors contributed to the discussion of the results. The manuscript was prepared by AS and RF and reviewed by all authors.

**Funding** This report is independent research funded by the National Institute for Health Research (Research for Patient Benefit programme, PB-PG-0815-20024). This study is based in part on data from the Clinical Practice Research Datalink obtained under licence from the UK Medicines and Healthcare products Regulatory Agency.

**Disclaimer** The views expressed in this publication are those of the authors and not necessarily those of the NIHR or the Department of Health and Social Care.

**Competing interests** None declared.

**Patient consent for publication** Not required.

**Provenance and peer review** Not commissioned; externally peer reviewed.

**Data availability statement** Data are available upon reasonable request. Availability of data and materials: The data that support the findings of this study are available from CPRD but restrictions apply to the availability of these data, which were used under license for the current study, and so are not publicly available. The datasets extracted from CPRD and used in the analysis during the current study are available from the corresponding author on reasonable request and consistent with the permission received from CPRD.

**ORCID iDs**
Rory Ferguson http://orcid.org/0000-0002-3256-6434
Daniel Prieto-Alhambra http://orcid.org/0000-0002-3950-6346
George Peat http://orcid.org/0000-0002-9008-0184
Antonella Delmestri http://orcid.org/0000-0003-0388-3403
Kelvin P Jordan http://orcid.org/0000-0003-4748-5335
Jose Maria Valderas http://orcid.org/0000-0002-9299-1555
Dahai Yu http://orcid.org/0000-0002-8449-7725
Alan Silman http://orcid.org/0000-0001-8426-8925

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
