## [Reviewer comments · BMJ Open]

ARTICLE DETAILS

TITLE (PROVISIONAL)	Does pre-existing morbidity influence risks and benefits of total hip replacement for osteoarthritis: a prospective study of 6682 patients from linked national datasets in England?
AUTHORS	Ferguson, Rory; Prieto-Alhambra, Daniel; Peat, George; Delmestri, Antonella; Jordan, Kelvin; Strauss, Vicky; Valderas, Jose; Walker, Christine; Yu, Dahai; Glyn-Jones, Sion; Silman, Alan

VERSION 1 – REVIEW

REVIEWER	Kuperman, EF Department of Internal Medicine, University of Iowa, Carver College of Medicine
REVIEW RETURNED	31-Dec-2020

GENERAL COMMENTS	Major concerns: 1. The relationship between morbidities and patient adverse events appears to be more complex than the abstract and the text of the results section implies. For example, prescriptions and PCP contacts do not appear to have significantly different ORs. The current abstract text for results implies a stronger relationship than observed in this study. This is slightly clearer from the results section of the manuscript, but again is oversimplified in discussion and conclusions implying a clearer relationship than observed in the data. I would advise spending more time justifying your summary statements or revising the summary statements to more accurately reflect a complex relationship incompletely predicted by the measured variables. Minor concerns: 1. Given the general audience of this journal, I would recommend some additional description of what a significant changes in OHS and EQ-5D are so that the significant observed benefits of surgery are better appreciated by the reader.2. The impact of morbidities on individual outcomes is not described in sufficient detail. For instance, the length of stay is described only in a vague and incomplete manner. Even if this would not fit within the main body of the manuscript, it would be reasonable to include this information within the supplementary material or (alternatively) excise from the paper altogether.3. A THA is expected to cause durable health benefits far in excess of the 90-day follow-up period. Reflection on the long-term impact of hip surgery would be appropriate for the discussion of this paper and if no data is available for this patient sample it would be reasonable to list as a limitation of this study.
--

REVIEWER	Cnudde, Peter Swedish Hip Arthroplasty Register
REVIEW RETURNED	04-Jan-2021

GENERAL COMMENTS	Thank you for a very interesting study. I did like the use of the different measures for comorbidity as predictive measures? There will always be the discussion whether or not this is a prospective study? Some purists will be adamant that this study cannot be called prospective as the data was not collected with the research question in mind, register researchers will claim the opposite. I am a little afraid that the first 2 references do not make sense at all and are very broad and not really covering the substance. The abstract does miss punctuation and are wrong with regards to PROMs (small s) and EQ-5D (misses the -). The increased mortality as a result of the increased comorbidity has been described previously and am really unsure whether this paper adds new information on 90-day mortality. Especially the rather wide CI of the OR (inclusive of 1) and the limited number of events might make the answer statistically wobbly. The question of HRQoL has been partially answered by Green et al in her work titled Comorbidity does not predict THA outcome. The use of the change in EQ-5D/OHS might be a little more novel and I agree besides the fact that your HRQoL/PROMs data is only on approx 20% of the study population, it shows the difference in gains are clinically comparable independent of most of the comorbidity measurement tools. I am a little at loss with regards to the complications. They seem to be rather lower than the historically recorded ones and likely to be related to quality of the HES data. Would it not be better to study the LOS and the readmissions as a proxy for complications (see Martinelli et al.), as you have that data anyway. There is available data on BMI and social deprivation, why did not use this in your adjustment? I do follow the rationale as stated in the discussion, but why mention it in the first instance. HES data is only available for England and as such the data is only relevant for England (not the UK!). Whilst the increased risk are modest (0.7 to 2.3%), this might not be considered modest for some patients, please explain. The gains are similar, but might not take the more frail patient to a similar level than the less frail. Is the absolute 6-month postop figure clinically different. I do believe this paper and the data has got potential but it could benefit from a more thorough use of the data and I would like to see a little more attention to detail. The conclusion could benefit to be worked out better and more systematically.
---

REVIEWER	Wilkinson, J. Mark Sheffield Teaching Hosp NHS Fdn Trust
REVIEW RETURNED	17-Jan-2021

GENERAL COMMENTS	This is an interesting paper that is essentially a follow on analysis of the sub-cohort of the patients described in the companion article bmjopen-2020-046713 entitled "Influence of pre-existing multimorbidity on receiving a hip arthroplasty: Cohort study of 28,026 elderly subjects from UK primary care", with which is it best read in conjunction.
---

	Here, the authors focus on at the outcomes for that subcohort of CPRD GOLD patients who do go on to THA surgery, with outcomes stratified by comorbidity status. Similar to its companion paper, perhaps the most novel association is that between frailty index and outcome. The other findings are largely confirmatory in nature but add to the body of evidence, as the data sources used here complement similar findings already in the public domain from secondary care datasets. What is really missing form this manuscript is to very clearly state in the introduction and setting that this data is a subset of the other study to give the clear context for the reader. If read in isolation, then these critical interpretation elements are lost. The obvious limitations of this datasubset are somewhat different to the primary care work and need to be expanded upon. These include the issue of confounding by indication, and the selective well-patient effect of undergoing surgery that impacts interpretation. An obvious missed oportunity is to examine the the 60% of patients with hip OA in primary care who didn't undergo hip surgery and compare morality rates. Presumably this data is available from their overall dataset and would be very helpful to include in a revised submission. Such an analysis would also help to clarify the impact of overall mortality risk versus benefit THA surgery and its rehabilitation has on patients, stratified by morbidity score. We might see that although the surgery has a small increment of risk, the overall benefits on mobility and health outweigh the former (notwithstanding the limitations of subtle selection biases). Or perhaps not. Either way, this is an obvious question to address given the data already collected.
--	---

VERSION 1 – AUTHOR RESPONSE

Authors' general comment:

1. The manuscript has been extensively revised and hopefully addresses all the concerns raised.
2. As part of our revisions we have added 2 Supplementary Tables and an additional text Appendix. We believe, both in terms of manuscript length and, also based on the reviewers' comments these are best placed outside the main manuscript but this is obviously an editorial decision
3. As this is part of a two paper companion set, we have referenced the companion paper as reference 17 (submitted)

1. Reviewer: 1: Major concerns:

1.1. *The relationship between morbidities and patient adverse events appears to be more complex than the abstract and the text of the results section implies. For example, prescriptions and PCP contacts do not appear to have significantly different ORs. The current abstract text for results implies a stronger relationship than observed in this study. This is slightly clearer from the results section of the manuscript, but again is oversimplified in discussion and conclusions implying a clearer relationship than observed in the data. I would advise spending more time justifying your summary*

statements or revising the summary statements to more accurately reflect a complex relationship incompletely predicted by the measured variables.

We are happy to address this issue in relation to the different parts of the manuscript. The differences highlighted by the reviewer only refer to the relation between health status and risk, whereas there were no obvious differences between the measures assessed and benefits. We have made the following changes:

Abstract: It is a challenge to keep the abstract concise given the number of different predictors and outcomes. We do feel that the summary was fair but have added a conditional sentence to the abstract (Page 3, lines 31-32).

Results: It is always intended that the reader should read the tables and our experience is that most readers and journals do not prefer to have too detailed a narrative in the results section when the data should speak for themselves. However, we have added some additional sentences to the results giving additional highlighting to some of the differences referred to (Page 8, line 24; Page 8, line 41)

Discussion: We have added an enlarged opening paragraph to the discussion (Page 10, lines 1-8)

Minor concerns

1.2. *Given the general audience of this journal, I would recommend some additional description of what a significant changes in OHS and EQ-5D are so that the significant observed benefits of surgery are better appreciated by the reader.*

We acknowledge that although some readers will be very familiar with these outcome measures and their interpretation, others are not. Mindful of the need to keep the word limit within the bounds set, we have added this information as an Appendix (Appendix) and referred the readers to this in our results section, (Page 9, line 20; page 9 line 37).

1.3. *The impact of morbidities on individual outcomes is not described in sufficient detail. For instance, the length of stay is described only in a vague and incomplete manner. Even if this would not fit within the main body of the manuscript, it would be reasonable to include this information within the supplementary material or (alternatively) excise from the paper altogether.*

We agree but were aware of adding to what was already a 'table heavy' paper. Thanks therefore for suggesting this as readers might find this useful. We had indeed done this analysis and have now added a paragraph to the results (Page 9, lines 4-12) and the data added as a Supplementary Table 3 (Page 28)

1.4. *A THA is expected to cause durable health benefits far in excess of the 90-day follow-up period. Reflection on the long-term impact of hip surgery would be appropriate for the discussion of this paper and if no data is available for this patient sample it would be reasonable to list as a limitation of this study.*

The benefits were collected at 6 months not 90 days which is already stated in the paper. Indeed 6 months is probably the minimum given the delay to completion in some patients (Page 7, lines 1-4). We acknowledge the reviewer's point about impact beyond this period is not without its own challenge, given other health issues that might intervene in any longer interval. We have added this to the discussion (Page 11, lines 27-30)

2. Reviewer: 2

2.1. *Thank you for a very interesting study. I did like the use of the different measures for comorbidity as predictive measures? There will always be the discussion whether or not this is a prospective study? Some purists will be adamant that this study cannot be called prospective as the data was not collected with the research question in mind, register researchers will claim the opposite. Absolutely agree with this point and changed to 'longitudinal record linkage study' (Page 5, line 34)*

2.2. *I am a little afraid that the first 2 references do not make sense at all and are very broad and not really covering the substance.*

We can see the Reviewer's point here. We have re-written these sentences to address the lack of clarity (page 5, lines 2-4)

2.3. *The abstract does miss punctuation and are wrong with regards to PROMs (small s) and EQ-5D (misses the -).*

Amended

2.4. *The increased mortality as a result of the increased comorbidity has been described previously and am really unsure whether this paper adds new information on 90-day mortality. Especially the rather wide CI of the OR (inclusive of 1) and the limited number of events might make the answer statistically wobbly.*

Mortality was not the major focus of the study but the Reviewer is right that we should emphasise the previous connection which we have in the discussion with additional references (Page 13, lines 5-60). We also agree the wide confidence intervals limit the conclusion of this aspect and this is added to the results description (Page 8, lines 35-37). The new information is the range of measures of multimorbidity – both for this and the other outcomes.

2.5. *The question of HRQoL has been partially answered by Green et al in her work titled Comorbidity does not predict THA outcome. The use of the change in EQ-5D/OHS might be a little more novel and I agree besides the fact that your HRQoL/PROMs data is only on approx 20% of the study population, it shows the difference in gains are clinically comparable independent of most of the comorbidity measurement tools.*

We had referred to Greene et al work (our reference 12). I think this is a comment on the paper and not a criticism requiring amendment

2.6. *I am a little at loss with regards to the complications. They seem to be rather lower than the historically recorded ones and likely to be related to quality of the HES data. Would it not be better to study the LOS and the readmissions as a proxy for complications (see Martinelli et al.), as you have that data anyway.*

The LOS data are referred to by Reviewer 1 above and we have added this as above. We do not disagree with the reviewer and that we had expected higher rates of complications which is why we undertook the comprehensive literature review to give some external validity to these findings (Page 10, lines (30-40).

Using readmission as a proxy for complications is an interesting suggestion. It cuts both ways of course, and many, rightly would have been critical if we had used this in preference to our direct measure. However, we had done this analysis and are happy to introduce the data (Page 9, lines 4-9; Supplementary Table 2). We return to this point in the discussion (Page 10 lines 1-5), where we indicate that with some variation between the different pre-operative measures, the two sets of data: direct complication rates and readmission rates are mainly similar in the size of their effect.

2.7. *There is available data on BMI and social deprivation, why did not use this in your adjustment? I do follow the rationale as stated in the discussion, but why mention it in the first instance.*

We are pleased that the Reviewer followed our rationale and thus it probably makes sense not to mention these 'in the first instance', and this has been deleted from the methods (see Page 11). We have kept the reference in the discussion to these as non-epidemiological readers hopefully would also agree with our reasoning

2.8. *HES data is only available for England and as such the data is only relevant for England (not the UK!).*

Changed where wrongly quoted, mainly in Tables

2.9. *Whilst the increased risk are modest (0.7 to 2.3%), this might not be considered modest for some patients, please explain. The gains are similar, but might not take the more frail patient to a similar level than the less frail. Is the absolute 6-month postop figure clinically different.*
This comment about 'modest' interestingly emerged from our consultation with a panel of patients at the end of the study but of course this was their subjective view and hence not appropriate for inclusion in this submission. We have modified our conclusions on risk (Page 10, lines 1-5)

2.10. *I do believe this paper and the data has got potential but it could benefit from a more thorough use of the data and I would like to see a little more attention to detail. The conclusion could benefit to be worked out better and more systematically.*
Obviously this is a general comment but we trust that the additional material and points referred to by this and the other reviewers addresses the points about 'more thorough use of the data (eg our new supplementary tables) as well as the other changes'

3. Reviewer: 3

3.1. *This is an interesting paper that is essentially a follow on analysis of the sub-cohort of the patients described in the companion article bmjopen-2020-046713 entitled "Influence of pre-existing multimorbidity on receiving a hip arthroplasty: Cohort study of 28,026 elderly subjects from UK primary care", with which is it best read in conjunction.*

Here, the authors focus on at the outcomes for that subcohort of CPRD GOLD patients who do go on to THA surgery, with outcomes stratified by comorbidity status. Similar to its companion paper, perhaps the most novel association is that between frailty index and outcome. The other findings are largely confirmatory in nature but add to the body of evidence, as the data sources used here complement similar findings already in the public domain from secondary care datasets.

What is really missing from this manuscript is to very clearly state in the introduction and setting that this data is a subset of the other study to give the clear context for the reader. If read in isolation, then these critical interpretation elements are lost

The first 2 paragraphs are more narrative than criticism of the work.

We appreciate the point in the third paragraph is well taken. Our challenge was that if the papers were not going to be published as companion pieces then each needed to stand alone. On consideration we think the Reviewer is correct that our providing the headline from the companion paper, established the basis more clearly for this submission. We have therefore added to the introduction two sentences (Page 5, lines 16-20)

3.2. *. The obvious limitations of this data-subset are somewhat different to the primary care work and need to be expanded upon. These include the issue of confounding by indication, and the selective well-patient effect of undergoing surgery that impacts interpretation.*
There are limitations most particularly the unknown issue as to why some patients with similar levels of multimorbidity have, and others don't have surgery. The reviewer is right as this is an obvious consequence of the observational nature of this analysis. Of course, there would never be a clinical trial that would randomise patients with multimorbidity to surgery or no-surgery. We had made the general point but have added a specific potential example of unmeasured confounding to raise the wider challenge (Page 11, lines 37-41)

3.3. *An obvious missed opportunity is to examine the 60% of patients with hip OA in primary care who didn't undergo hip surgery and compare mortality rates. Presumably this data is available from their overall dataset and would be very helpful to include in a revised submission.*

Such an analysis would also help to clarify the impact of overall mortality risk versus benefit THA surgery and its rehabilitation has on patients, stratified by morbidity score. We might see that although the surgery has a small increment of risk, the overall benefits on mobility and health outweigh the former (notwithstanding the limitations of subtle selection biases). Or perhaps not. Either way, this is an obvious question to address given the data already collected.

The hypothesis behind this point is that there would be a number of expected deaths in the general population of this age and hence the observed mortality in an operated cohort should be considered against the expected mortality rate. Indeed, the reviewer is proposing a morbidity standardised mortality risk, to properly assess any reported increased post-surgical mortality. This makes sense!

The Reviewer also tacitly refers to other data showing the long term benefit to health from hip surgery and that allowing for baseline morbidity, despite such a standardisation exercise as above, even if there was a short term post-operative mortality hit, THA has a long term effect on survival.

These great questions are outside the scope of this manuscript. Firstly, our focus was on very short term mortality (90 days) and that is too short a window, notwithstanding the small number of deaths, to make any useful comment on the longer term benefit. Secondly mortality was not a major focus of this paper. Our interest in risks was much more on complications and hospitalisations. Similarly, our benefits were also on a much shorter time scale.

(For the this Reviewer's benefit, we did look at the data and 90 days was too short a window for any useful conclusion)

VERSION 2 – REVIEW

REVIEWER	Kuperman, EF Department of Internal Medicine, University of Iowa, Carver College of Medicine
REVIEW RETURNED	01-Apr-2021

GENERAL COMMENTS	I think this draft represents as significant improvement, and I believe the conclusion that comorbidities as estimated by the 5 methods of the authors do not represent contraindications to elective surgery is well-supported from the results. However, there are still some areas that I believe can be further strengthened prior to publication: Major concerns:  1. While I appreciate that the nuance of the results are difficult to fit within the confines of abstract word count, I don't believe that the additional sentence to the results section is specific enough to add value to the manuscript. The Charlson CI, and (especially) # of PC contacts do not significant predict 90-day medical complications in Table 4 and only EFI and (possibly) number of comorbid diseases appears to predict mortality in Table 5, with all other confidence intervals overlapping 1. The statement in the abstract that peri-operative complication rate and mortality "both increased with worse pre-operative health status" is not supported by the results of this manuscript. Likewise, the abstract conclusions are not supported by the results of this paper. 2. Likewise, within the body of the paper I have concerns that the results section does not accurately reflect the nuance of the observed data, although the added word count somewhat addresses. For medical complications, I would advise excising general statements such as "These results were similar result [sic] across the different approaches to assessing multimorbidity." PC
---

	contacts did not perform well, and Charlson CI only slightly better. It would be just as fair to generalize that these measures performed poorly in predicting complications, even though this is an unexpected finding. The poor predictive value represents a strength towards your overall conclusion (comorbidity does not equal contraindication) and could be highlighted in your results. Minor concerns: 1. The grammar and syntax of abstract and paper still need attention. The results section is especially difficult to parse in its current construction. Examples include: the second sentence of abstract-results reverse to patients with "more adverse health[,] as well as the duplicated word highlighted above.
--	--

REVIEWER	Cnudde, Peter Swedish Hip Arthroplasty Register
REVIEW RETURNED	01-Apr-2021

GENERAL COMMENTS	Certainly a further improvement and I would be happy to support acceptance, pending some further clarifications/adjustments. However there remain some issues with sloppiness and some questions remain on the originality as well as the message of this manuscript. The impact of poor health on higher mortality is well known, the novelty is looking at perhaps number of prescriptions (how many of those are analgesia?)/GP visits/what is the definition of a GP visit (is a phone/mail for a repeat prescription classed as a contact/visit)? How do eFI, GP visits, CCI, prescriptions and comorbidities relate? More importantly how do they relate with the (preop) EQ-5D as this a patient's view on their QoL. The fact that poorer entry levels of PROMs are associated with poorer postop PROMs, despite the fact that gains might be similar, is also known. Some style remarks: Punctuation after sentences is not consistent (likely to be picked up later-is this to fulfil character requirements?). See point at first review. Issues remain. page 8/29 not or no? page 9/36 does not make sense "for primary care contacts...." please reread/adjust. Is the conclusion really valid? Impact of preoperative health measures (which ones?-why not looking at EQ-5D and OHS) on the risks (death/LOS/readmissions/complications/AE)...and benefits (EQ-5D/PROMs) Also... the association between health status, using EQ-5D, and mortality was previously published by Cnudde et al. IJERPH, 2017, 14, 899. Is preoperative patient-reported health status associated with mortality after THA? It might be worthwhile reading the conclusions as they do provide a not dissimilar message.
---

REVIEWER	Wilkinson, J. Mark Sheffield Teaching Hosp NHS Fdn Trust
REVIEW RETURNED	28-Mar-2021

GENERAL COMMENTS	My previous queries have now been addressed.
--

VERSION 2 – AUTHOR RESPONSE

Reviewer 1: Major concerns

1. Abstract

The Reviewer in his opening paragraph concludes our conclusions are 'well supported from the results'. His major concern was that the abstract-as he acknowledges is constrained by the word count – does not reflect accurately what we found. We accept that with such a large body of data, as in this ms, with 5 key predictors and several outcomes, there is a challenge within a 300 word count limit as to which results are highlighted and how the conclusions are framed. We are happy to change the abstract to reflect the emphasis suggested by the Reviewer (our page 3, lines 26-28, 33-345).

2. Results

The reviewer does not criticise our data as presented in the tables but suggest our narrative of one of the outcomes - the number of complications – needed to be tightened and made more readable. We have therefore heavily edited those relevant paragraphs of the result section (our page 8, line 33 to page 9, line 11). Indeed, we welcome and acknowledge that this more nuanced narrative provides support for our overall conclusion that any concern about risks is more than compensated by the benefits from surgery. We have therefore added a sentence to this effect to the discussion (our page 10, lines 31-32)

3. Minor concerns

We have changed “more adverse health” to “worse pre-operative health” (our page 3, line 30). We have carefully gone over the ms to see if we can improve the grammar where this was needed and the changes can be seen in the tracked changes version, including the rewriting of the results paragraphs as above

Reviewer 2

1. “However there remain some issues with sloppiness”

We feel this is an inappropriately loaded comment for a reviewer to make without good grounds. He has given no evidence that the study was not performed to the highest standards. His concerns apart from some missing full stops relate to his view of the originality and that measures of morbidity extracted from the primary care record are inferior to a one-off patient reported assessment of their general health status.

2. Some questions remain on the originality as well as the message of this manuscript. The impact of poor health on higher mortality is well known

The reviewer, as he is perfectly entitled to do questions the originality of this work, with reference to his own previous publications. Without attempting to get into a competitive situation, our response is that our present study is so different in scope and breadth that its originality remains and that our use of the cumulative medical record does not negate his work on patient self-report QoL but rather adds substantially to it. We noted that the reviewer refers to his paper: (Cnudde et al, Int J Environ Res and Pub Health 2017 14 899-912) . This work focused on patient self-reported Quality of Life (using EQ-

5D-3L) measured at the time of surgery and its association with mortality 5 years later. We also note The second study, that he indirectly refers to (first author Bulow et al, Act Orthop 2017 88 472-477) which was similar in its aim construct with the single outcome being long term mortality. The predictors in this latter paper were restricted to the Charlson and the closely related Elixhauser indices. By contrast to these papers, our current study looked at several outcomes in terms of short term hazards and also benefits in the same analysis. We also had multiple measures of health status as predictors. Although the current study is therefore very different in scope and breadth from Dr Cnudde's work, we acknowledge that we could have given greater attention and emphasis on the difference from our current work. Thus, we are happy to quote these work in the introduction - now cited as Reference 19 (our page 5, line 16) and refer again to this work in our discussion (References 19 and 35)

3. How do eFI, GP visits, CCI, prescriptions and comorbidities relate?

A detailed statistical analysis of the statistical concordance and relationship between these different constructs of health is beyond the scope of this ms. There is clearly not a single summary statistic to measure 'health' and we are not sure what such an analysis would bring apart from adding to what is an already large set of tables. More importantly the aim of the paper was to show that routine data can be used to derive a number of different accepted constructs of health, and there was no existing 'best buy'. We had originally added a short statement on the potential for overlap but have enlarged this (our page 11, lines 34-38)

4. More importantly how do they relate with the (preop) EQ-5D as this a patient's view on their QoL.

Firstly, although we agree that EQ-5D provides a patient view of their health, the focus of this paper was on establishing the relationship between measures of health that could be derived from routine primary care data. EQ-5D is not routinely collected and also is subject to variation. Thus, these are separate questions. We have added a statement referring to this point and making mention of this Reviewer's publication -new Reference 35 (our Page 11, line 40 to page 12, line 4) emphasising the separate importance of how patients' own perceptions of their health can influence outcome but this is addressing a different question.

5. The fact that poorer entry levels of PROMs are associated with poorer postop PROMs, despite the fact that gains might be similar, is also known.

We don't disagree but that was not the point of the analysis. We have shown in Tables 7 and 8 that the gains are similar across all the measures of pre-operative health status we have measured and this had not been studied previously

6. Some style remarks:

Punctuation after sentences is not

Thank you for your forensic reading! We have added some missing full stops.

page 8/29 not or no?

amended (Our page 8, line 23)

page 9/36 does not make sense "for primary care contacts...." please reread/adjust.

Done (our page 9 lines3,6)

7. Is the conclusion really valid? Impact of preoperative health measures (which ones?-why not looking at EQ-5D and OHS) on the risks (death/LOS/readmissions/complications/AE)...and benefits (EQ-5D/PROMs)

We have explained above we have focused on using health measures that can be derived from routine primary care collected data. The use of patient derived EQ5D and OHS as predictors of outcome, both risk and benefit, address a separate question as we have answered above. We have added a short section on this (our page 10, line 40 to page 11, line 4)

8. Also... the association between health status, using EQ-5D, and mortality was previously published by Cnudde et al. IJERPH, 2017, 14, 899. We have answered this point above and referenced Dr Cnudde's paper (new reference 35)

VERSION 3 – REVIEW

REVIEWER	Kuperman, EF Department of Internal Medicine, University of Iowa, Carver College of Medicine
REVIEW RETURNED	18-Jun-2021
GENERAL COMMENTS	I am satisfied with the authors responses to my suggestions on the previous draft.